# Multigene Panel Germline Testing of 1333 Czech Patients with Ovarian Cancer

**DOI:** 10.3390/cancers12040956

**Published:** 2020-04-13

**Authors:** Klara Lhotova, Lenka Stolarova, Petra Zemankova, Michal Vocka, Marketa Janatova, Marianna Borecka, Marta Cerna, Sandra Jelinkova, Jan Kral, Zuzana Volkova, Marketa Urbanova, Petra Kleiblova, Eva Machackova, Lenka Foretova, Jana Hazova, Petra Vasickova, Filip Lhota, Monika Koudova, Leona Cerna, Spiros Tavandzis, Jana Indrakova, Lucie Hruskova, Marcela Kosarova, Radek Vrtel, Viktor Stranecky, Stanislav Kmoch, Michal Zikan, Libor Macurek, Zdenek Kleibl, Jana Soukupova

**Affiliations:** 1Institute of Biochemistry and Experimental Oncology, First Faculty of Medicine, Charles University, 128 53 Prague, Czech Republic; Klara.Lhotova@lf1.cuni.cz (K.L.); lenka.stolarova@lf1.cuni.cz (L.S.); petra.boudova@lf1.cuni.cz (P.Z.); mjana@lf1.cuni.cz (M.J.); boreckam@gmail.com (M.B.); marta.cerna@lf1.cuni.cz (M.C.); sandra.jelinkova@lf1.cuni.cz (S.J.); jan.kral@lf1.cuni.cz (J.K.); zuzana.klusonova@lf1.cuni.cz (Z.V.); zdekleje@lf1.cuni.cz (Z.K.); 2Institute of Biology and Medical Genetics, First Faculty of Medicine, Charles University and General University Hospital in Prague, 128 00 Prague, Czech Republic; Marketa.Urbanova@vfn.cz (M.U.); pekleje@lf1.cuni.cz (P.K.); 3Department of Oncology, First Faculty of Medicine, Charles University and General University Hospital in Prague, 128 08 Prague, Czech Republic; Michal.Vocka@vfn.cz; 4Department of Cancer Epidemiology and Genetics, Masaryk Memorial Cancer Institute, 656 53 Brno, Czech Republic; emachack@mou.cz (E.M.); foretova@mou.cz (L.F.); hazova@mou.cz (J.H.); vasickova@mou.cz (P.V.); 5Department of Medical Genetics, Centre for Medical Genetics and Reproductive Medicine, Gennet, 170 00 Prague, Czech Republic; Filip.Lhota@gennet.cz (F.L.); Monika.Koudova@gennet.cz (M.K.); Leona.Cerna@gennet.cz (L.C.); 6Department of Medical Genetics, AGEL Laboratories, AGEL Research and Training Institute, 741 01 Novy Jicin, Czech Republic; spiros.tavandzis@lab.agel.cz (S.T.); jana.indrakova@lab.agel.cz (J.I.); 7Department of Medical Genetics, GHC Genetics, 110 00 Prague, Czech Republic; hruskova@ghc.cz; 8Department of Medical Genetics, Pronatal, 147 00 Prague, Czech Republic; kosarova@pronatal.cz; 9Department of Medical Genetics, University Hospital Olomouc, Faculty of Medicine and Dentistry, Palacky University Olomouc, 779 00 Olomouc, Czech Republic; radek.vrtel@fnol.cz; 10Research Unit for Rare Diseases, Department of Pediatrics and Adolescent Medicine, First Faculty of Medicine, Charles University and General University Hospital in Prague, 12808 Prague, Czech Republic; vstra@lf1.cuni.cz (V.S.); skmoch@lf1.cuni.cz (S.K.); 11Department of Gynecology and Obstetrics, Hospital Na Bulovce and First Faculty of Medicine, Charles University, 180 81 Prague, Czech Republic; michal.zikan@lf1.cuni.cz; 12Laboratory of Cancer Cell Biology, Institute of Molecular Genetics of the Czech Academy of Sciences, 142 20 Prague, Czech Republic; libor.macurek@img.cas.cz

**Keywords:** ovarian cancer, next-generation sequencing, predisposition genes, cancer risk, mutation

## Abstract

Ovarian cancer (OC) is the deadliest gynecologic malignancy with a substantial proportion of hereditary cases and a frequent association with breast cancer (BC). Genetic testing facilitates treatment and preventive strategies reducing OC mortality in mutation carriers. However, the prevalence of germline mutations varies among populations and many rarely mutated OC predisposition genes remain to be identified. We aimed to analyze 219 genes in 1333 Czech OC patients and 2278 population-matched controls using next-generation sequencing. We revealed germline mutations in 18 OC/BC predisposition genes in 32.0% of patients and in 2.5% of controls. Mutations in *BRCA1/BRCA2*, *RAD51C/RAD51D*, *BARD1*, and mismatch repair genes conferred high OC risk (OR > 5). Mutations in *BRIP1* and *NBN* were associated with moderate risk (both OR = 3.5). *BRCA1/2* mutations dominated in almost all clinicopathological subgroups including sporadic borderline tumors of ovary (BTO). Analysis of remaining 201 genes revealed somatic mosaics in *PPM1D* and germline mutations in *SHPRH* and *NAT1* associating with a high/moderate OC risk significantly; however, further studies are warranted to delineate their contribution to OC development in other populations. Our findings demonstrate the high proportion of patients with hereditary OC in Slavic population justifying genetic testing in all patients with OC, including BTO.

## 1. Introduction

Ovarian cancer (OC) is the most severe gynecologic malignancy with stable incidence and mortality. The most frequent OC types (85–95%) are epithelial tumors, which are high-grade (HG) serous in 70% of cases [1,2]. Because of the nonspecific symptoms and a lack of presymptomatic screening modalities, most women are diagnosed with an advanced disease, having a dismal 25% 5-year survival rate [3].

The overall OC lifetime risk oscillates around 2% in the general female population in developed countries. Central and Eastern Europe, including the Czech Republic, represented a region with the highest OC incidence (11.9 ASRW per 100,000 females) and mortality (6.0 ASRW per 100,000 females) worldwide in 2018 (http://gco.iarc.fr). In the Czech Republic alone, annual OC incidence and mortality in 2018 reached 9.5 and 6.7 ASRW per 100,000 females, respectively.

Genetic predisposition for OC is unusually high and is reported in up to 25% of cases [4,5,6]. The most frequent germline mutations affect the *BRCA1* and *BRCA2* genes, conferring 24% and 8.4% OC lifetime risks, respectively [7]. The *BRCA1* and *BRCA2* mutation carriers frequently but not exclusively develop HG serous OC [8]. Carriers of mutations in these major OC predisposition genes have also very high risk of breast cancer (BC) development. A high OC risk has also been associated with germline mutations in *RAD51C, RAD51D,* Lynch syndrome genes, and *STK11*; a moderate OC risk with *BRIP1* [9,10,11,12,13]. Risks associated with germline mutations in genes with anticipated BC and/or OC predisposition (incl. *ATM, BARD1, CDH1, CHEK2, NBN, PALB2*, *PTEN,* and *TP53*) and in other candidate genes remain to be determined [14,15,16,17]. The identification of presymptomatic women at high risk who can benefit from risk-reducing salpingo-oophorectomy (RRSO) is of critical importance, as demonstrated by the reduced OC mortality in *BRCA1* and *BRCA2* mutation carriers undergoing preventive surgery [18].

In this report, we aim to establish an association of germline mutations with OC in the Czech patients belonging to the Slavic population that has not been systematically analyzed for OC predisposition. Seven Czech genetic laboratories participated in the analysis of 1333 Czech OC patients by the identical procedure using CZECANCA panel (CZEch CAncer paNel for Clinical Application) targeting 219 genes [19]. Prevalence of variants in genes affected in OC patients was assessed in 2278 population-matched controls. This analysis enabled us to comprehensively determine mutations frequency and clinicopathological characteristics of OC in carriers of mutations in genes with known OC predisposition but also to analyze contribution of population-specific variants in other candidate genes to OC predisposition.

## 2. Results

### 2.1. Description of Study Population

Altogether, samples obtained from 1333 OC patients diagnosed at seven centers were analyzed by the identical panel NGS using the CZECANCA panel targeting 219 cancer-predisposition and candidate genes and were evaluated centrally by the identical bioinformatics pipeline. From 1333 analyzed OC patients, 1045 (78.4%) women were diagnosed with OC only and 288 (21.6%) women with double primary tumors, including BC (210 patients; 15.8%) or other tumors (78 patients; 5.9%). The median age at OC diagnosis was 53.7 years (range 15–86 years). Almost half (47.6%) of the patients had a negative family cancer history. From 1120 OC patients with known histology, 728 (65.0%) women developed serous adenocarcinoma with prevailing HG tumors. Sixty percent of cases represented patients with advanced disease (stages III–IV). The clinicopathological characteristics are provided in Appendix A.

### 2.2. Mutations in 18 Known/Anticipated Hereditary BC/OC Genes

We primarily focused on mutations in 18 BC/OC genes listed in the NCCN Guidelines for Genetic/Familial High-Risk Assessment: Breast, Ovarian, and Pancreatic (Version 1.2020; 4 December 2019). We identified 441 mutations in 427/1333 (32.0%) OC patients and 58/2278 (2.5%) mutation carriers among population-matched controls (PMC) in 18 known/anticipated BC/OC genes (Figure 1, Table 1, and Appendix A). Thirteen multiple mutation carriers (Figure 1) identified among patients only (characterized in Appendix A) were excluded from the subsequent analyses.

Carriers of germline mutations in 10 genes (including Lynch syndrome genes analyzed as a group together) had significantly increased OC risk (Table 1 in bold). We found the prevailing *BRCA1* or *BRCA2* germline alterations in 323/1320 (24.5%) patients and in 12/2278 (0.5%) PMC. Further, 65/1320 (4.9%) OC patients carried a mutation in 8 other genes significantly associated with OC risk in our study (including 2 carriers of mutations in *STK11*, an established high-risk OC gene that did not reach significant association in our study due to low frequency of mutation carriers in patients; Figure 1). We found only 19/2278 (0.8%) carriers of mutations in these 8 genes in PMC. 

The copy number variation (CNV) analysis in 18 OC/BC genes revealed 37 large genomic rearrangements in 37/1333 (2.8%) patients. They affected seven genes (23×*BRCA1*, 4×*BRIP1*, 4×*CHEK2*, 2×*MLH1*, 2×*STK11*, 1×*PALB2*, and 1×*CDH1*) and accounted for 8.4% (37/441) of all pathogenic mutations in these genes. Except 1 whole gene duplication of *MSH6* (classified as VUS), we found no CNV in analyzed controls in these 18 genes.

### 2.3. Clinical and Histopathological Characteristics of Mutation Carriers

Subsequently, we described the clinicopathological characteristics of the mutation carriers in 10 genes associated with OC risk (Figure 2 and Appendix A). Multiple mutation carriers (Appendix A) were excluded from this analysis.

#### 2.3.1. Age at OC Diagnosis

The highest mutation frequency was found in patients diagnosed with OC at 40–49 and 50–59 years (37.4% and 40.7%, respectively) and the lowest in patients diagnosed before the age of 30 (8.3%; Figure 2A). Interestingly, the mutation frequency in the group of the oldest patients (≥70 years) was twice higher than in the youngest (<30 years) patients’ subgroup (*p* = 0.013 for difference). This difference was primarily caused by *BRCA1*/*BRCA2* mutations (3.6% vs. 18.1% in patients <30 vs. ≥70 years), as the frequency of *non-BRCA* genes mutations was similar (4.8% vs. 4.3%). The median age at diagnosis was significantly different in *BRCA1* (51.0 years; range 23–78) and *BRCA2* (58.4 years; range 27–78) mutation carriers (*p* = 8.5×10^−10^), respectively. The median age at diagnosis in other genes with at least 10 identified mutation carriers increased gradually from *RAD51C* (52.2 years; range 25–69) to *NBN* (54.5 years; range 18–76), *RAD51D* (56.0 years; range 36–69), and *BRIP1* (58.0 years; range 30–71). We observed a younger median age at diagnosis in carriers of mutations in Lynch syndrome genes 46.0 years (range 35–73).

#### 2.3.2. Personal and Family Cancer History

The highest proportion of mutations (109/203; 53.7%) was detected in double primary OC and BC patients, while in patients diagnosed with OC only and double primary OC and non-BC cancer, it reached 256/1038 (24.7%) and 21/79 (26.6%), respectively (Figure 2B). The frequency of mutations in patients from hereditary OC families (HOC) was 49.1% (57/116; Figure 2C). Decreasing proportion of mutation carriers in other family cancer history categories (41.0% in HBOC and 29.4% in multiple cancer) was dominantly caused by decreasing *BRCA1* mutation prevalence. Nevertheless, in 587 OC patients without a positive family cancer history, we still identified 120 (20.4%) carriers of pathogenic mutations.

#### 2.3.3. Stage and Histology

Almost 60% of patients were diagnosed at FIGO stage III or IV (Figure 2D). In contrast, 6/8 informative Lynch syndrome gene mutation carriers were diagnosed with stage I tumors.

The mutation rate stratified OC into two histological clusters. The high mutation rate subgroup included 879 patients with HG/unspecified serous, borderline, and endometrioid tumors with 303 (34.5%) carriers, while the low mutation rate subgroup included 232 patients with low-grade (LG) serous, mucinous, clear cell, and other tumors with 28 (12.1%) carriers. *BRCA1/2* mutations in HG serous carcinomas were more than twice as frequent (146/472; 30.9%) as in LG serous ones (11/84; 13.1%). Interestingly, the distribution of *BRIP1/RAD51C*/*RAD51D* mutations among histological types was similar to that of *BRCA1/2*. The lowest proportion of mutations (7/90; 7.8%) was found in rare histological cancer types (herein denominated as “Other”).

### 2.4. Mutations in Additional 201 Analyzed Genes

Finally, we reviewed the presence of germline variants in additional 201 genes targeted by the CZECANCA panel [19]. This analysis revealed 230 mutations in 89 genes in 208 (15.6%) patients (Appendix A). Of these, 149 (11.2%) patients carried mutations in “additional” genes exclusively while 59 (4.4%) patients carried a mutation in “additional” genes alongside a mutation in one of the 10 OC risk genes. Mutations in these “additional” genes were rare and their prevalence was significantly higher in patients over controls in only four genes (Table 2). However, only mutations in *PPM1D* were significantly associated with OC risk (*p* = 0.003) following Bonferroni correction and exclusion of carriers of mutations in OC predisposition genes. All *PPM1D* mutations were mosaic with MAF = 14%–60% and MAF = 17%–19% in patients and controls, respectively. It should be noted that blood for genetic testing was sampled after the application of chemotherapy in all *PPM1D* positive patients (in average at 38 months after treatment; ranged 4 months–7.1 years). Seven out of 15 *PPM1D* mutation carriers harbored an additional mutation in another DNA repair gene (3×*BRCA2*, 1×*PALB2*, 1×*EXO1*, and 1×*PMS1*). MAF of *PPM1D* mutations correlated neither with age at OC diagnosis nor with the time from the last chemotherapy (Appendix A). Mutations in *PPM1D* and *SHPRH* were significantly associated only with age > 60 years (*p* = 0.001), whereas frequency of *NAT1* mutations in particular categories was similar (Appendix A). Uncorrected p values were marginally significant also for germline variants in *MMP8* and *FANCG* in OC patients when carriers of mutations in 10 BC/OC predisposition genes significantly associating with OC risk in our study were excluded (Table 2).

## 3. Discussion

The analysis of 1333 Czech OC patients and 2278 population-matched controls provides the most comprehensive view of the genetic architecture of OC predisposition in the Slavic population. From 18 OC/BC predisposition genes listed in current NCCN breast/ovarian familial cancer guidelines, mutations in 10 genes were significantly associated with OC risk in our population being present in 399/1333 (29.9%) OC patients and 31/2278 (1.4%) PMC (Figure 1). Mutations in eight remaining genes were extremely rare (*CDH1, PTEN, STK11,* and *TP53*) or absent (*CDKN2A* and *NF1*) or did not significantly differ in frequency among cases and controls (*ATM, PALB2,* and *CHEK2*). Mutations in *BRCA1/2, RAD51C/D,* and Lynch syndrome genes were associated with a high OC risk, while mutations in *BRIP1* were associated with a moderate OC risk in our study (Table 1), in concordance with previous reports [9,10,20,21]. The *BRCA1* and *BRCA2* mutations, present in 84.0% of all mutation carriers, were by far the most frequent alterations found in 17.9% and 7.4% of our patients, respectively. Mutations in other eight genes leaded by *RAD51C/RAD51D/BRIP1* affected additional 5.0% of patients, as shown also by others recently [5,6,22]. Germline mutations in Lynch syndrome genes together associated with high OC risk. Mutations in *MLH1* prevailed similarly as in Lynch syndrome patients diagnosed with colorectal cancer [23].

In contrast to previous studies, our results suggest increased OC risk in carriers of *NBN* and *BARD1* mutations [12,24]. We did not find significant increase of OC risk for carriers of mutations in *ATM* and *PALB2*, as noticed previously [12,24,25]. However, further analyses considering very large population-matched studies or studies considering families of mutation carriers can better disclose moderate risk associations, as shown for *PALB2* mutations recently [26].

Overrepresentation of mutations in the *CHEK2* gene in OC patients in this study was marginally nonsignificant in contrast to our previous report where we identified moderately increased OC risk for *CHEK2* mutation carriers [27]. However, last four *CHEK2* coding exons were not targeted in our gene panel omitting possible deleterious *CHEK2* alterations identified in our previous study in which last four coding exons were analyzed separately in both cases and controls. Mutations in *NF1* were absent and were extremely rare in *CDH1* and *PTEN*, just like *STK11* mutations found in a patient with nonepithelial OC, a characteristic Peutz–Jeghers syndrome manifestation [9]. Altogether, the high overall frequency of mutations in OC predisposition genes in our study is in agreement with some previous studies [4,5,6,28] and may contribute to a high OC incidence in our population.

Multigene testing revealed 13 carriers of multiple pathogenic mutations (1.0% of patients). Similar frequency of individuals with this multilocus inherited neoplasia alleles syndrome (MINAS) [29] was shown also in previous analyses of OC patients [30,31].

We analyzed available phenotype characteristics in 1320 OC patients with one pathogenic mutation at the most in 10 genes associated with OC risk in our study (Figure 2). While the highest prevalence of *BRCA1/2* mutation carriers was in patients diagnosed with double primary OC and BC, mutations in *RAD51C/RAD51D/BRIP1* prevailed in patients diagnosed with OC only (Figure 2B); nevertheless, their distribution among histological subtypes was similar to that in *BRCA1/2* mutation carriers (Figure 2E). In contrast to Castera et al. who found mutations in *RAD51C/RAD51D/BRIP1* dominantly in French OC patients with a positive family OC history [32], we identified mutations in these genes in 1/116 (0.9%) and 22/587 (3.7%) carriers in HOC patients and in patients with a negative family cancer history, respectively. Further, we have noticed a surprisingly high frequency of OC-predisposing mutations in older patients. Their prevalence in patients ≥ 60 years was 23.6%, whereas Harter et al. found in this age group 18.9% mutation carriers even though frequency of mutation carriers in patients <60 years in both studies was comparable (32.6% and 33.2%, respectively) [28]. *BRCA1* mutations dominated in patients <60 years over *BRCA2* mutations, while in patients ≥ 60 years, their frequencies were comparable. Moreover, we revealed 29 *BRCA1/2* mutation carriers (13.9% of patients) in 208 OC patients diagnosed at ≥60 years with no family cancer history, while Morgan and colleagues detected only two (4.3%) *BRCA1/2* mutations in 46 sporadic OC patients ≥ 60 years [33]. Even in the oldest subgroup of our OC patients diagnosed at ≥70 years, the frequency of *BRCA1/2* mutation carriers exceeded 18%, while in other studies, *BRCA1/2* mutations’ frequency in this age category was below 10% [34,35]. This high frequency of *BRCA1/2* mutations in our patients ≥70 years contrasted with a low frequency in women diagnosed at <30 years (18.1% vs. 3.6%; *p* = 0.003; Figure 2A). The difference was even more apparent in “sporadic” OC cases (with no family cancer history), where *BRCA1/2* mutations were found in 6 out of 45 (13.3%) women ≥70 years but in none of 52 cases diagnosed at <30 years. It should be emphasized that although rare histological OC types were more frequent in the subgroup of 52 patients diagnosed with sporadic OC at <30 years, 32 (65.3%) of 49 informative cases developed invasive epithelial OC.

Mutations in OC predisposition genes significantly prevailed in subgroups with high-grade/ nonspecified serous, borderline, and endometrioid tumors over subgroup with low-grade serous, mucinous, clear cell, or other rare histologic types (Figure 2E). Surprisingly, the overall mutation frequency in patients with borderline tumors was comparable with that of in HG serous OC (32.2% and 36.7%, respectively; Figure 2E). Thus, we compared mutation frequency in patients with no family cancer history diagnosed with these histological tumor types, and we found that although the mutation frequency in sporadic borderline tumors was half in comparison to sporadic HG serous (Figure 3), it still largely exceeded 10% in both hereditary and sporadic cases, justifying the genetic testing of borderline tumors. The large proportion of borderline tumors with positive family cancer history in our study suggested that this OC subtypes belong to a possible manifestation of a cancer predisposition. However, our observation needs to be confirmed in other populations as current reports about borderline tumors in *BRCA1/2* mutation carriers are limited.

The multigene panel enabled us to identify other candidate genes associating with increased OC risk. We noticed many rare truncating variants episodically affecting various genes and clustering into *PPM1D*, *NAT1*, and *SHPRH* in OC patients. The *PPM1D* gene, coding for WIP1 phosphatase, was the only candidate associated with OC risk following multiple testing correction. Similarly to the previous studies describing its mosaic variants in OC patients [36,37,38], we also found mosaic gain-of-function mutations resulting in increased WIP1 phosphatase activity [38]. All *PPM1D* mutations in our patients were identified in postchemotherapy treatment blood samples suggesting their somatic origin [39]. Germline mutations in *NAT1* have not been analyzed for OC predisposition so far. However, several polymorphisms in *NAT1* (coding for arylamine N-acetyltransferase 1 engaged in carcinogen metabolism and detoxification) were shown to modify the risk of various cancers [40,41]. The *SHPRH* gene codes for E3 ubiquitin-protein ligase targeting PCNA upon DNA damage [42]. Contribution of *SHPRH* germline variants to OC risk remains elusive. Overall, low mutation frequencies found in gene candidates in our study precluded its precise OC risk estimations and will require large, multiethnic, case-control studies, segregation analyses in affected families, and functional analyses. Alongside variants clustering to a few candidate genes, we identified rare mutations in a gene family coding for Fanconi anemia (FA) proteins involved in the repair of DNA interstrand crosslinks [43]. Several FA genes belong to established OC predisposition genes, including *BRCA1* (*FANCS*), *BRCA2 (FANCD1*), *RAD51C (FANCO)*, *PALB2 (FANCN)*, and *BRIP1 (FANCJ).* Except these, we found rare mutations in other FA genes (*FANCG*, *FANCD2*, and *FANCA*) in 11 (0.83%) of 1333 OC patients compared to 5 in 2278 PMC (0.2%), with cumulative OR = 3.78 (95% CI 1.21–13.91; *p* = 0.02). Interestingly, these rare mutations were detected almost exclusively in patients without mutations in other OC predisposition genes.

The strengths of this study include an identical NGS analysis and bioinformatics pipeline in all patients, a careful curation of clinical data, and an ethnically homogeneous set of patients and controls representing the largest sample set from the region of Central and Eastern Europe. Despite that, the number of individuals still did not allow the precise OC risk calculations in rarely mutated genes. Although all OC cases in the Czech Republic are eligible for genetic testing, OC patients with positive family cancer history and earlier-onset individuals were enriched in our study, especially in a small subgroup enrolled before 2015 (in the Center A only).

Whether the high prevalence of clinically important germline mutations in OC patients justifies population-wide screening is a vivid matter of debate [44,45,46,47,48]. We emphasize that we found *BRCA1/2* mutations in 14.5% of OC patients with no family cancer history who would currently not be revealed presymptomatically without population screening. We assume that careful application of germline testing in all OC patients and their relatives would reduce OC burden in our population. Moreover, the mutations in *BRCA1/2* [49,50] and other OC predisposition genes [51,52] represent valuable predictive biomarkers improving OC chemotherapy.

## 4. Materials and Methods

Analyzed patients (*N* = 1333) were enrolled in 2010–2018 and included all OC cases regardless of familial cancer history or OC histology subtypes. As knowledge about germline mutations’ frequency in women diagnosed with BTO is limited, we included these histological subtypes to our study. Clinicopathological data were obtained during genetic counselling or retrieved from the patients’ records. OC patients with a positive cancer family history were stratified into (i) hereditary ovarian cancer (HOC) families with OC and other nonbreast cancer (BC) in the family history; (ii) hereditary breast and/or ovarian cancer (HBOC) families with BC and OC or other cancer in the family history, and (iii) multiple cancer families with non-OC and non-BC in the family history. Index patients were tested in seven centers: (A) First Faculty of Medicine, Charles University, Prague (*N* = 637); (B) Masaryk Memorial Cancer Institute, Brno (*N* = 357); (C) Gennet, Prague (*N* = 273); (D) AGEL Laboratories, Novy Jicin (*N* = 34); (E) GHC Genetics (*N* = 12); (F) Pronatal (*N* = 11), and (G) University Hospital Olomouc (*N* = 9). 

Population-matched controls (PMC; *N* = 2278) included 616 noncancer controls collected in centers A (*N* = 344), B (*N* = 150), and D (*N* = 122), and 1662 unselected controls provided by the National Center for Medical Genomics (http://ncmg.cz). The noncancer controls were volunteers (78 males and 538 females) aged ≥ 60 years without a personal or family cancer history (in first-degree relatives). The unselected controls (1170 males and 492 females; median age 57 years, range 18–88 years) were unrelated individuals analyzed by whole-exome sequencing (WES) for various noncancer conditions.

All patients and controls were Caucasians of a Czech origin. Written informed consent was obtained from all patients and controls. The study was approved by the Ethics Committee of the General University Hospital in Prague; ethics approval number was 92/14. The study was performed in accordance with the Declaration of Helsinki.

### 4.1. Next-Generation Sequencing

Germline blood-derived DNA was analyzed by the CZECANCA (CZEch CAncer paNel for Clinical Application; custom-made SeqCap EZ choice panel; Roche) panel NGS targeting 219 genes on MiSeq (Illumina), as described in details previously [19]. Sequencing reads were aligned by Novoalign v2.08.03 to the human reference genome (hg19). Variants were identified using GATK and Pindel, CNVs using CNV score [19]. The entire diagnostic pipeline was successfully tested using European Molecular Genetics Quality Network schemes (EMQN) and validated as we have described recently [19]. 

### 4.2. Variant Classification

We first analyzed 18 genes considered clinically relevant to the HBOC syndrome (MIM #604370) by NCCN, namely, *ATM, BARD1, BRCA1, BRCA2, BRIP1, CDH1, CHEK2, MLH1, MSH2, MSH6, NBN, PALB2, PTEN, RAD50, RAD51C, RAD51D, STK11,* and *TP53.* Germline variants (with frequency ≤ 0.01 and ≤0.05 in 1000 Genomes project and noncancer PMC, respectively) were classified into three groups: i) pathogenic/likely pathogenic, ii) variants of unknown significance (VUS), and iii) likely benign/benign, based on recommendations from the ENIGMA consortium (https://enigmaconsortium.org). All nonsense/frameshift/splicing (± 1–2 bp) mutations/CNVs were considered pathogenic/likely pathogenic unless classified as other in the ClinVar database; whole gene duplications were considered VUS. The other types of mutations were considered pathogenic/likely pathogenic only if classified as such in ClinVar by at least two submitters. TP53 variants were classified using the IARC TP53 database (http://p53.iarc.fr/), *CHEK2* VUS using a recently published functional assay [27].

Subsequently, we analyzed variants in another 201 genes targeted by the CZECANCA panel. Nonsense/frameshift/splicing (± 1–2 bp) mutations/CNVs (except whole gene duplications) with frequency ≤0.01 and ≤0.05 in 1000 Genomes project and in noncancer PMC, respectively, were considered pathogenic. 

All pathogenic/likely pathogenic mutations in patients and noncancer PMC were confirmed by Sanger sequencing and CNVs by MLPA (if available) or by qPCR (protocol available on request), and they were submitted to ClinVar under the submission ID SUB5822876.

### 4.3. Statistical Analysis

The odds ratio (OR) for particular gene was calculated using Fisher’s exact test, and p values <0.05 were considered significant. The multiple mutation carriers were excluded from the OR calculations. For the identification of other OC candidate genes, the Bonferroni correction was employed. The associations between mutation status and clinicopathological characteristics were estimated using Fisher’s exact test, and p values <0.05 were considered significant.

## 5. Conclusions

Our study demonstrated that nearly one in three OC patients carries a pathogenic mutation in genes significantly associated with OC. The mutation frequency exceeded 10% in all clinicopathological subgroups, regardless of the age at diagnosis, clinical or histopathological characteristics, with an exception of women diagnosed with OC before the age of 30 or with rare histological OC subtypes. Importantly, we found that the high mutation prevalence included borderline tumors justifying genetic testing of all OC patients, including women diagnosed with borderline tumors. Surprisingly, *BRCA1/2* mutations were not associated with sporadic OC in very young women (≤30 years). Besides the established OC predisposition genes, *NBN* and *BARD1* were significantly associated with a moderate OC risk; however, further studies will be required to specify the associated OC risk and to identify the value of the detected genetic mutations in terms of disease prognosis and therapy prediction. Hence, analyses of rarely mutated BC/OC predisposition genes that failed to increase OC risk in our study are further warranted to evaluate their association with OC in future larger dataset and/or in frame of international consortia. These should include also other candidate alterations with predictive and/or prognostic potential.

## Figures and Tables

**Figure 1 cancers-12-00956-f001:**
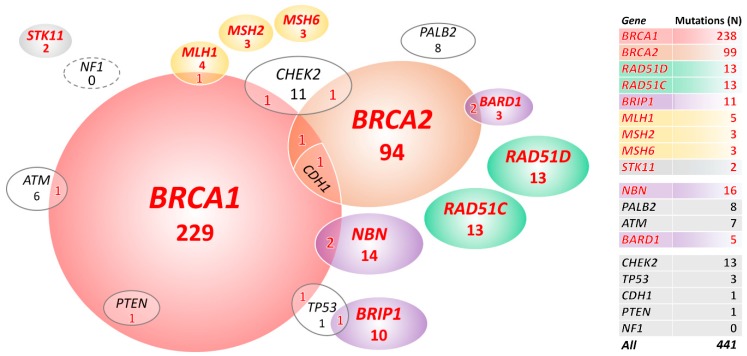
Overall, 427 mutation carriers of 441 mutations in 18 known/anticipated breast cancer (BC)/ovarian cancer (OC) predisposition genes. In total, 399 carriers in genes significantly associated with OC in our study are highlighted in red letters. *STK11* is highlighted as rarely mutated but established OC predisposition gene.

**Figure 2 cancers-12-00956-f002:**
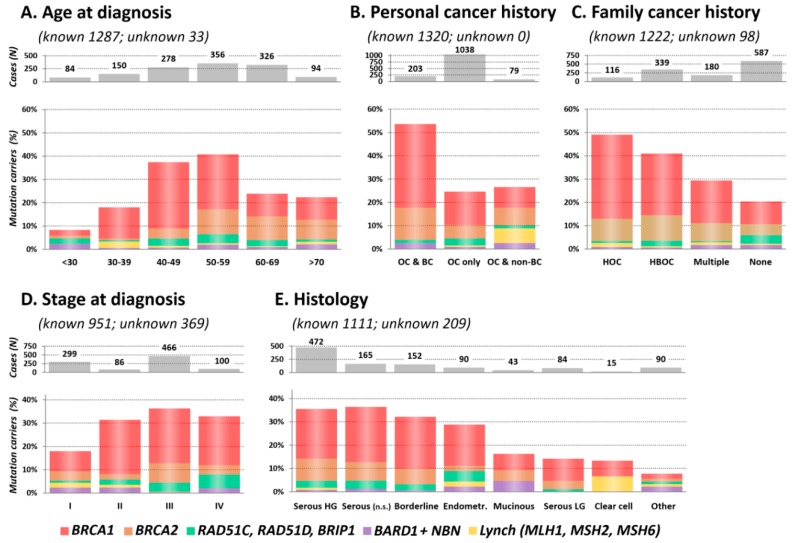
Proportion of mutation carriers in clinicopathological subgroups, including (**A**) Age at OC diagnosis; (**B**) Personal cancer history; (**C**) Family cancer history; (**D**) Stage at diagnosis; (**E**) Histology in 1320 OC patients.

**Figure 3 cancers-12-00956-f003:**
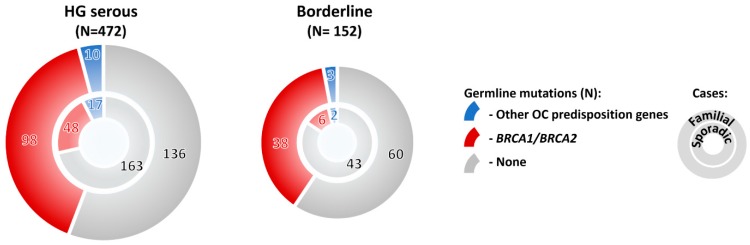
Frequency of mutations in 10 BC/OC predisposition genes significantly associated with OC in our study in OC patients with high-grade (HG) serous and borderline tumors, respectively. The patients were subdivided into subgroups with positive (familial cases) and negative (sporadic cases) family cancer history, respectively.

**Table 1 cancers-12-00956-t001:** Mutation frequencies in 1320 ovarian cancer cases and in 2278 population-matched controls (PMC).

Gene	1320 OC Patients ^(a)^ N Mutations (%)	2278 PMC N mutations (%)	OR (95% CI); p ^(a)^
Increased OC risk ^(b)^			
***BRCA1 ^(^*^c)^**	**229 (17.35)**	**5 (0.22)**	**95.2 (40.1–295.2); 1.83 × 10^−97^**
***BRCA2 ^(^*^c)^**	**94 (7.12)**	**7 (0.31)**	**24.9 (11.6–63.6); 1.16 × 10^−33^**
***RAD51D***	**13 (0.98)**	**2 (0.09)**	**11.3 (2.6–103.4); 9.66 × 10^−5^**
***RAD51C***	**13 (0.98)**	**4 (0.18)**	**5.7 (1.7–23.8); 0.001**
***BRIP1 ^(c)^***	**10 (0.76)**	**5 (0.22)**	**3.5 (1.1–13); 0.03**
***MLH1 ^(c)^***	**4 (0.3)**	**1 (0.04)**	6.9 (0.7–340.4); 0.06 ^(d)^
***MSH2***	**3 (0.23)**	**0**	**0.049 ^(d)^**
***MSH6***	**3 (0.23)**	**0**	**0.049 ^(d)^**
*STK11*	2 (0.15)	0	0.13
Potentially increase or insufficient evidence OC risk ^(b)^			
***NBN ^(c)^***	**14 (1.06)**	**7 (0.31)**	**3.5 (1.3–10.2); 0.006**
*PALB2*	8 (0.61)	9 (0.40)	1.5 (0.5–4.5); 0.45
*ATM ^(c)^*	6 (0.45)	8 (0.35)	1.3 (0.4–4.3); 0.78
***BARD1 ^(c)^***	**3 (0.23)**	**0**	**0.049**
No increased risk of OC ^(b)^			
*CHEK2 ^(c)^*	11 (0.83)	8 (0.35)	2.4 (0.9–6.8); 0.06
*TP53* ^(c)^	1 (0.08)	2 (0.09)	0.9 (0–16.6); 1
*CDH1* ^(c)^	0	0	-
*PTEN* ^(c)^	0	0	-
*NF1*	0	0	-

^(a)^ Prevalence of mutations in all 1333 patients (including 13 multiple mutation carriers) is provided in Appendix A. ^(b)^ Gene classification according to the NCCN guidelines version 2020.1. ^(c)^ Excluding 13 multiple mutation carriers described in Figure 1 and Appendix A. ^(d)^ When analyzed Lynch syndrome genes collectively: OR = 22.63 (95% CI 3.4–958.5); *p* = 1.95 × 10^−05^.

**Table 2 cancers-12-00956-t002:** Additional 201 analyzed genes significantly associated with OC risk in the group of all OC patients and in a subgroup of 934 patients without mutations in 10 established OC predisposition genes.

Gene	Patients N Mutations (%)	2278 PMC N Mutations (%)	OR (95% CI); p (Bonferroni Corrected *p*)
All 1333 OC patients			
***PPM1D***	16 (1.20)	2 (0.09)	13.82 (3.24–124.22); 7.4 × 10^−6^ (0.001)
***NAT1***	13 (0.98)	5 (0.22)	4.48 (1.49–16.07); 0.003 (n.s.)
***SHPRH***	5 (0.38)	1 (0.04)	8.57 (0.96–404.83); 0.028 (n.s.)
934 OC patients without mutations in 10 genes significantly associated with OC in our study			
***PPM1D***	12 (1.28)	2 (0.09)	14.80 (3.28–136.67); 1.7 × 10^−5^ (0.003)
***NAT1***	8 (0.86)	5 (0.22)	3.96 (1.13–15.30); 0.026 (n.s.)
***MMP8***	6 (0.64)	4 (0.18)	3.67 (0.87–17.74); 0.041 (n.s.)
***FANCG***	5 (0.53)	2 (0.09)	6.12 (1.00–64.45); 0.025 (n.s.)

n.s., nonsignificant.

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
