# Peer review of "Multigene Panel Germline Testing of 1333 Czech Patients with Ovarian Cancer"

_cancers, 2020, doi:10.3390/cancers12040956_

Round 1
Reviewer 1 Report
This paper comprehensively showed the results of oncogenic panel tests in Czech ovarian cancer patients.However, more discussion would be needed to support the results.
1. What is the rationale to obtain borderline tumors? Please add more rationale and the results reported in other studies for readers.
2. Please mention how statistical analysis was perdformed in materal & methods.
3. Did you perform MLPA to validate large genomic rearrangements?
4. In this study, male contols were included in PMC. I am concerned that this will affect the calculation of the odds ratio of ovarian cancer. Why are men included in the analysis? Isn't it necessary to proceed with the analysis except men?
Reviewer 2 Report
In this manuscript Lhotova et al present a genomic profiling of a huge Czech cohort of ovarian cancer patients. The data was compared to a population-matched control cohort and they show a high prevalence of germline mutations. On the whole the manuscript is well presented and includes interesting findings. There are however some data analysis issues and the manuscript requires modification before it could be acceptable for publication.
Abstract
A clear aim for the study is missing, and the text should be better focused
Introduction
The text can shorten and better focused and structured
- I will suggest the following structure:
- General information about ovarian cancer including prognosis, symptoms and subtypes
- Prevalence and genetic predisposition
- What can the genetic information be used for (prevention, selection of treatment)?
- Aim
I will suggest adding /changing the following elements:
- Why is the incidence in the Czech Republic so high? (This should also be the focus for the Discussion)
- Most breast cancers are sporadic, only a minor fraction is caused by inheritance of pathogenic germline variants
Material and methods
- Try to reduce the number of repetitions in the text.
- Start with the description of the cohorts. The information about the laboratories can come later
- Why where males used as controls?
Results
The text can be shortened, many of the paragraphs can be merged.
Discussion
Should be shorten and more focused
- Select 3 or 4 main findings and discuss them. Do not discuss all the findings
- The second part: The focus should be more on the biological relevance and clinical consequences of the findings (prevention and selection of treatment)
